# Effects of biological sex mismatch on neural progenitor cell transplantation for spinal cord injury in mice

Michael Pitonak[1], Miriam Aceves[1,2], Prakruthi Amar Kumar [1], Gabrielle Dampf [1], Peyton Green[3], Ashley Tucker[1,2], Valerie Dietz[1], Diego Miranda[1], Sunjay Letchuman [1,4], Michelle M. Jonika [1,5], David Bautista[1], Heath Blackmon [1] & Jennifer N. Dulin [1,2] ✉

Despite advancement of neural progenitor cell transplantation to spinal cord injury clinical trials, there remains a lack of understanding of how biological sex of transplanted cells influences outcomes after transplantation. To address this, we transplanted GFP-expressing sex-matched, sex-mismatched, or mixed donor cells into sites of spinal cord injury in adult male and female mice. Biological sex of the donor cells does not influence graft neuron density, glial differentiation, formation of the reactive glial cell border, or graft axon outgrowth. However, male grafts in female hosts feature extensive hypervascularization accompanied by increased vascular diameter and perivascular cell density. We show greater T-cell infiltration within male-to-female grafts than other graft types. Together, these findings indicate a biological sex-specific immune response of female mice to male donor cells. Our work suggests that biological sex should be considered in the design of future clinical trials for cell transplantation in human injury.

Spinal cord injury (SCI) is a devastating event that typically results in lifelong neurological dysfunction due to permanent loss of neural tissue and disruption of spinal cord neural circuits. Transplantation of neural progenitor cells (NPCs) provides the injured spinal cord with new neurons that can support the formation of novel neural relays between graft- and host neurons, potentially supporting recovery of lost neurological function[1,2]. Although NPC transplantation has been examined in SCI models since the 1980s[3–7], there remains an incomplete understanding of the biological factors that affect outcomes following transplantation. Factors such as graft survival, integration, neuronal and glial differentiation, and graft-derived axon extension are commonly assessed anatomical outcome measures in preclinical transplantation studies[8–20]. However, there have been relatively few studies of immunological-related outcomes following transplantation. This an important consideration that must

be addressed before clinical efficacy of NPC transplantation therapies can be achieved.

In recent years, sex as a biological variable has received increasing attention in biomedical research[21–23]. In preclinical SCI research, the biological sex of experimental animals is now recognized as a significant factor influencing outcomes such as inflammation, response to pharmacological treatments, and the degree of functional recovery achieved[24]. In preclinical SCI cell transplantation studies, both male and female rodents have been used as host animals[25]. However, the sex of donor cells is not typically reported. Clinical organ transplantation studies have shown that biological sex mismatch between donor tissue and the recipient can significantly increase the risk of transplant rejection and other adverse outcomes (Table 1). These findings raise the question of whether a mismatch between the sex of transplanted cells and host animals could represent a previously unappreciated

[1]Department of Biology, Texas A&M University, College Station, TX 77843, USA. [2]Texas A&M Institute for Neuroscience, Texas A&M University, College Station, TX 77843, USA. [3]Ganado High School, Ganado, TX 77962, USA. [4]Mays Business School, Texas A&M University, College Station, TX 77843, USA. [5]Genetics Interdisciplinary Program, Texas A&M University, College Station, TX 77843, USA. ✉e-mail: jdulin@bio.tamu.edu

**Table 1 | Summary of selected clinical research studies reporting adverse outcomes following transplantation of male donor tissue into female recipients**

| Transplant type | Study design | Outcomes |
|---|---|---|
| Corneal transplantation (penetrating keratoplasty)[43] | The effects of H-Y incompatibility on graft rejection were retrospectively analyzed for $n = 229$ adult patients | Graft survival was significantly lower in females receiving male corneas compared to all other groups (male to male, female to female, and female to male) |
| Pediatric heart transplantation[39] | Orthotopic cardiac allograft rejection was monitored for $n = 61$ infants and children (12 years old or younger) during the first year post-transplantation | Female recipients of male hearts had significantly more graft rejection episodes at 3 months- and 1 year post-transplantation than all other groups |
| Adult heart transplantation[41] | Rejection episodes following heart transplantation in $n = 174$ adult patients were monitored at 1 year after heart transplantation | Female recipients of male hearts had significantly lower creatinine clearance and significantly higher rejection episodes; male recipients of female hearts exhibited significantly lower 1-year survival |
| Liver transplantation[40] | The incidence of graft failure due to chronic rejection was retrospectively analyzed for $n = 423$ adult consecutive primary liver allograft recipients | Female recipients of male livers showed a significantly increased probability of chronic rejection |
| Lung transplantation[42] | Survival, time to acute allograft rejection, and time to development of obliterative bronchiolitis was retrospectively measured in a population of $n = 98$ adult lung transplant recipients | Female recipients of male donor lungs had a significantly reduced time to obliterative bronchiolitis diagnosis |
| Kidney transplantation[46] | Kidney function (estimated glomerular filtration rate), acute rejection, and 5-year graft survival were retrospectively analyzed in $n = 230$ adult kidney transplant recipients | Female recipients of male transplants had significantly reduced kidney function, significantly higher risk of an acute rejection episode, and significantly worse 5-year graft survival |
| Kidney transplantation[45] | Rates of graft survival and death-censored graft survival at 1 and 10 years post-transplantation were retrospectively assessed for $n = 195,516$ adult kidney transplant recipients | Female recipients of male kidneys had significantly increased risk of graft failure in the first year, and between 2 and 10 years, compared to all other groups |

source of variability in experimental SCI studies. We therefore sought to address this question in a preclinical SCI/transplantation study utilizing syngeneic mice, in which donor tissue and recipient subjects are both on a C57BL/6 inbred background, allowing us to isolate the effects of biological sex on immune-related outcomes.

Here, we take advantage of a previously published genotyping protocol to distinguish male from female mouse embryos and obtain spinal cord NPCs from donor embryos of defined sex. Four weeks following transplantation of either male-only, female-only, or mixed male and female NPCs into sites of cervical SCI in adult male or female mice, we performed histological analysis of anatomical outcomes including graft survival, neuronal and glial differentiation, axon extension, vascularization, and infiltration of phagocytic cells and lymphocytes.

## Results

### Sex of donor cells does not substantially influence graft differentiation or axon outgrowth

Visual distinction of male versus female mouse embryos is not possible during the stage of embryonic development used for NPC isolation in our studies (-E12.5). To obtain NPCs from either male or female donor spinal cord tissue, we collected GFP⁺ E12.5 mouse embryos from 9 different litters. We performed genetic sex determination by genotyping individual embryos for the gametologs *Rbm31x* and *Rbm31y* (Fig. 1a, b), as previously described[26]. This approach takes advantage of the divergent genetic sequences of the X-linked gene (269 bp) and the Y-linked gene (353 bp), allowing for the rapid identification of sex from donor embryonic tissue (Fig. 1b). We found that all litters contained both male and female embryos in varying proportions (Table 2). This suggests that typical NPC transplantation experiments, in which NPCs are obtained from entire litters of embryos and pooled prior to transplantation[17,20,27–30], probably utilize donor cells of both sexes. NPCs were pooled according to sex and transplanted into sites of acute cervical dorsal column SCI in host mice, such that donor cells were either sex-matched [male host with male graft (MM); female host with female graft (FF)], sex-mismatched [female host with male graft (FM); male host with female graft (MF)], or mixed [either male or female hosts grafted with 1:1 ratio of mixed male and female NPCs (MX and FX, respectively)] (Fig. 1a). Four weeks following transplantation, all grafts exhibited similar survival to previous studies[17], with complete filling of

the lesion site (Fig. 1c). Grafts contained neurons and astrocytes, similar to previous reports (Fig. 1d)[17,18,27,29,30]. This indicates that storing donor embryonic spinal cords on ice for a ~4-h period prior to NPC isolation (during the process of genotyping) did not adversely affect graft survival or differentiation.

We next analyzed the differentiation of graft-derived cells. All grafts that we assessed were densely populated with neurons (Fig. 2a). We compared the density of neurons in each treatment group to the neuronal density of intact cervical spinal cord gray matter in adult mice (Fig. 2b). The intact spinal cord gray matter contained ~97000 ± 2780 neurons/mm³. Across all experimental groups, we observed broad variation in graft neuronal density, but no significant main effect of treatment (Fig. 2b, Supplementary Table 1). We next quantified the amount of GFAP immunoreactivity within grafts and the host spinal cord tissue immediately surrounding the lesion/graft, where the astroglial border is located[31] (Fig. 2c–e). GFAP signal within grafts was about twice as intense as in the uninjured cervical spinal cord (Fig. 2d), but we did not detect any significant differences between treatment groups. Similarly, we did not detect inter-group differences in GFAP intensity within the surrounding host spinal cord (Fig. 2e). We also noted that an obvious reactive cell layer with heightened GFAP immunoreactivity was not clearly visible at the host/graft border (Fig. 2c). This observation is consistent with previous reports that the presence of NPC grafts reduces glial reactivity around the lesion[4,5,14].

We then quantified outgrowth of GFP⁺ graft-derived axons at 500-μm intervals within the host white matter rostral and caudal to the graft (Fig. 2f). Overall, the treatment group was a significant source of variation ($P = 0.0389$), with significant differences between groups only observed at a few distance intervals. We did not detect any strong trends in which one group consistently exhibited higher or lower axons counts than any other group. However, at some locations, significant differences were apparent. For example, the MM group featured greater numbers of GFP⁺ axons than the FM group at −3500 to −2500 μm intervals (Fig. 2f). Together, these data indicate that the biological sex of host and donor tissue does not significantly influence commonly reported outcomes including NPC graft differentiation into neurons or astrocytes, astroglial reactivity surrounding the graft, or axon outgrowth.

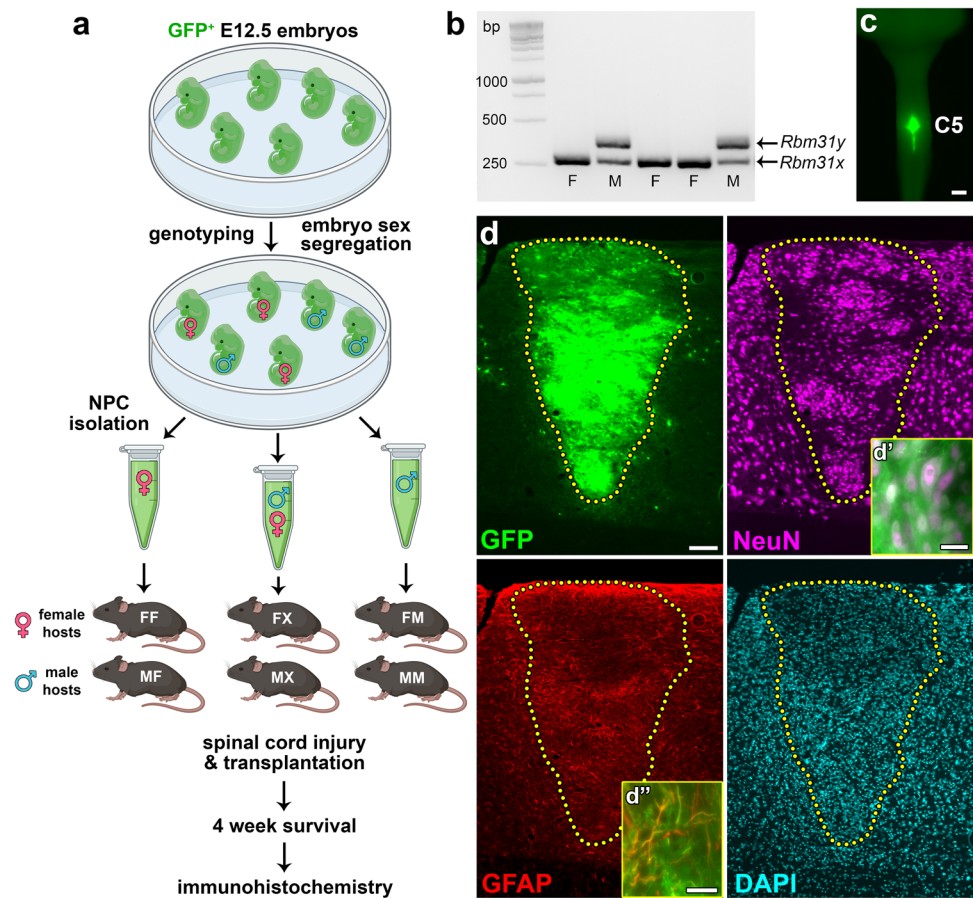

**Fig. 1 | Transplantation of neural progenitor cells of defined biological sex into sites of spinal cord injury. a** Experimental design. GFP⁺ E12.5 mouse embryos were genotyped to determine biological sex. Spinal cords from male or female embryos were pooled and neural progenitor cells (NPCs) were isolated. Male and female mice received spinal cord injury followed by immediate transplantation of either 100% female NPCs, mixed 50/50 male/female NPCs, or 100% male NPCs. Survival time was 4 weeks post-transplantation. FF female host, female graft, FX = female host, mixed graft, FM = female host, male graft, MF = male host, female graft, MX = male host, mixed graft, MM = male host, male graft. Image created with BioRender.com. **b** Genetic determination of embryo sex through PCR amplification of *Rbm31x* (269 bp) and *Rbm31y* (353 bp) gametologs. F female, M male. **c** Gross image of spinal cord explant containing a GFP⁺ NPC graft at spinal cord cervical level C5; dorsal view. **d** Representative image of an NPC graft at 4 weeks post-transplantation (female host with female graft). Host/graft boundary is indicated with a dotted line. GFP is expressed in all graft-derived cells; NeuN is expressed in all neurons; GFAP is expressed in astrocytes; DAPI labels cell nuclei. d', d" GFP⁺ graft-derived cells colocalize with neuron- and astrocyte-specific markers. Scale bars = 1 mm (**c**), 100 μm (**d**), 20 μm (d', d"). Source data are provided as a Source Data file. The experiments in **b** were performed twice with similar results. The experiments in panels c-d were performed three times with similar results.

## Male-to-female NPC grafts exhibit hypervascularization and perivascular hypercellularity

Vascularization of spinal cord NPC grafts is critical for transplant survival[32,33]. We have previously shown that grafts of dissociated spinal cord NPCs become vascularized after transplantation into sites of SCI[14,17], although it is not clear whether vascular structures within grafts are derived from host- and/or graft endothelial cells. Unexpectedly, we observed that grafts in the female host/male graft (FM) group exhibited atypical morphology, with large vascular-like structures present throughout graft tissue (Supplementary Fig. 1). To identify blood vessels within grafts, we performed immunolabeling against CD31 (PECAM-1), an antigen expressed on vascular endothelial cells[34] (Fig. 3a). CD31⁺ vascular structures did not express GFP, indicating that they are host-derived. Using CD31 immunoreactivity, we identified all blood vessels within grafts including transverse- and longitudinally oriented vessels (Fig. 3b). Quantification of the total vascular footprint in each graft type showed that female host/male graft (FM) subjects contained a significantly greater area occupied by vasculature within grafts compared to every other treatment group (Fig. 3c). Moreover, this group had a greater average vascular diameter than the MX, FF, and FX groups, indicating that blood vessels within FM grafts were enlarged (Fig. 3d, Supplementary Fig. 1). In addition, we observed hypercellularity within and around graft blood vessels, evident by increased density of DAPI⁺ cell nuclei in these regions (Fig. 3e). Because cell density was noticeably high in the space immediately surrounding graft blood vessels, we quantified cell density within the perivascular

## Table 2 | Sex of embryos from 9 individual litters used to obtain neural progenitor cells

| Litter | Total number of embryos | Male embryos | Female embryos |
|---|---|---|---|
| A | 7 | 4 (57.1%) | 3 (42.9%) |
| B | 7 | 3 (42.9%) | 4 (57.1%) |
| C | 7 | 5 (71.4%) | 2 (28.6%) |
| D | 8 | 6 (75%) | 2 (25%) |
| E | 8 | 3 (37.5%) | 5 (62.5%) |
| F | 8 | 4 (50%) | 4 (50%) |
| G | 9 | 6 (66.7%) | 3 (33.3%) |
| H | 11 | 6 (54.5%) | 5 (45.5%) |
| I | 12 | 5 (41.7%) | 7 (58.3%) |
| Overall | 77 | 42 (54.5%) | 35 (45.5%) |

Cells were obtained from heterozygous GFP⁺ E12.5 mouse embryos, product of *n* = 9 wild-type C57BL/6 female dams with *n* = 9 GFP⁺ male sires.

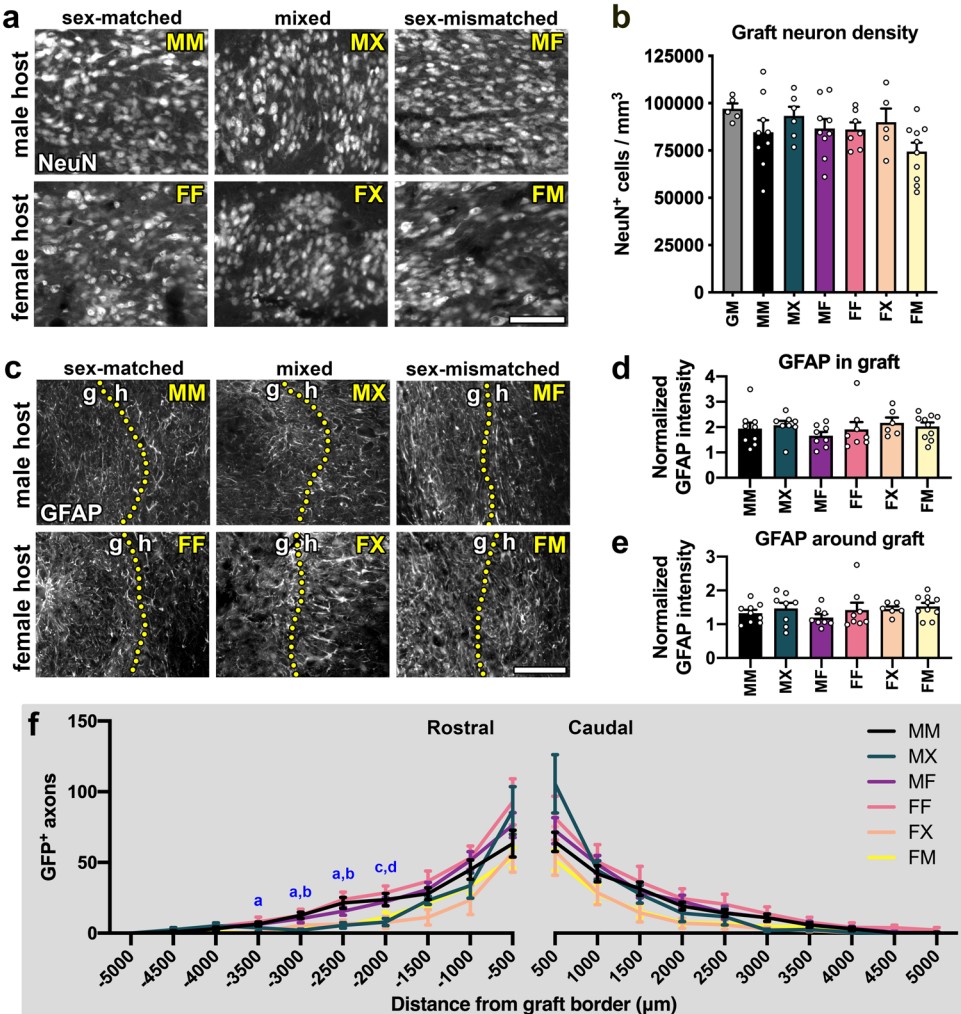

**Fig. 2 | Graft type does not significantly influence neuronal or glial differentiation. a** Representative images of graft-derived neurons (NeuN). **b** Quantification of graft neuron density in each experimental group and intact spinal cord gray matter (GM). *$P < 0.05$ vs. GM by ordinary one-way ANOVA with Dunnett's multiple comparisons test. Statistical analysis details are in Supplementary Table 1. GM ($n = 5$ mice), MM ($n = 9$ mice), MX ($n = 6$ mice), MF ($n = 9$ mice), FF ($n = 7$ mice), FX ($n = 5$ mice); FM ($n = 10$ mice). **c** Representative images of GFAP immunoreactivity in and around grafts. Dotted lines indicate the boundary between graft (**g**) and host (**h**) tissue. **d, e** Quantification of GFAP immunoreactivity (**d**) within the graft and **e** within a 250-µm interval of host tissue immediately surrounding the graft; data are normalized to GFAP intensity in intact spinal cord gray matter. No significant differences were detected by two-way ANOVA with

Sidak's multiple comparisons test. MM ($n = 9$ mice); MX ($n = 8$ mice); MF ($n = 8$ mice); FF ($n = 8$ mice); FX ($n = 6$ mice); FM ($n = 10$ mice). **f** Quantification of axon outgrowth within 30-µm tissue sections at 500-µm intervals rostral and caudal to the graft. MM ($n = 9$ mice), MX ($n = 5$ mice), MF ($n = 9$ mice), FF ($n = 8$ mice), FX ($n = 6$ mice), FM ($n = 10$ mice). All data are mean ± SEM. **a** $P < 0.05$ for MM vs. FM; **b** $P < 0.05$ for MM vs. MX; **c** $P < 0.05$ for MX vs. MF; **d** $P < 0.05$ for MX vs. FF by two-way repeated measures ANOVA with Tukey's multiple comparisons test. MM male host, male graft, MX male host, mixed graft, MF male host, female graft, FF female host, female graft, FX female host, mixed graft, FM female host, male graft. Scale bars = 100 µm. Source data are provided as a Source Data file. The experiments in panels a and c were performed twice with similar results.

space in a 50-µm radius (Fig. 3f). We found that this was significantly higher in the FM group compared to all other groups (Fig. 3g). Together, these findings reveal that female host/male graft subjects exhibit abnormal graft vascularization and cellular infiltration suggestive of an immune response.

## Increased infiltration of cytotoxic T cells in male grafts within female mice

Perivascular cellular infiltrates in the injured mouse spinal cord have previously been shown to correspond to regions of fibrosis containing lymphocytes, macrophages, and fibrocytes[35]. However, little has been reported about the infiltration of host immune cells into NPC grafts placed within sites of SCI. Due to our observation of increased perivascular cell density in FM grafts, we next investigated host cell infiltration into grafts. After SCI, macrophages and microglia infiltrate the injury site and contribute to secondary tissue damage into the chronic

phase of injury[36]. Because macrophages and activated microglia are difficult to distinguish from each other histologically[37], we performed immunolabeling for multiple immunohistochemical markers for macrophages/microglia including CD68 (ED1) and Iba1 (Supplementary Fig. 2, Fig. 4a). We did not detect any significant effects of graft type on the number of CD68+ cells in grafts ($P = 0.142$; Fig. 4b). We did identify a significant main effect of graft type on Iba1 immunoreactivity within grafts ($P = 0.0295$), with FM grafts containing higher Iba1+ cellular density than MF grafts ($P = 0.330$; Fig. 4c). We also observed that the morphology of phagocytic cells varied topographically within grafts and between subjects (Fig. 4a). For both of these markers, intersubject variability in the density of microglia/macrophages was high within individual treatment groups; hence, other factors besides sex are likely to have greater influence on phagocytic cell density in grafts.

Previous studies have demonstrated that female recipients are significantly more likely to reject solid organ transplants if the donor

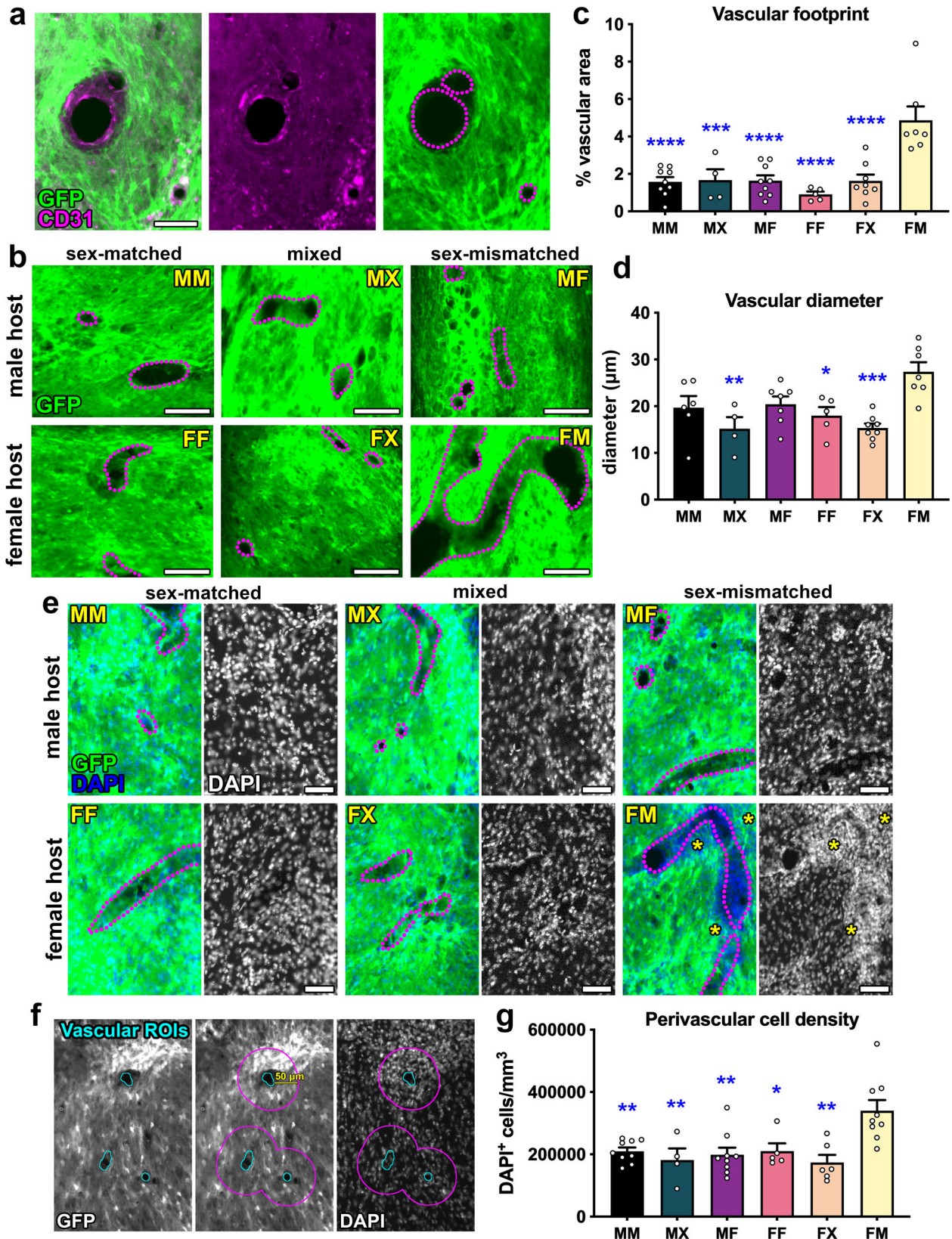

tissue is biologically male (Table 1)[38–46]. Immune rejection is mediated by host T lymphocytes (T cells), which recognize donor-derived cell surface antigens on the allograft tissue and infiltrate into graft tissue where they promote destruction of the graft[47]. We evaluated the presence of all T cells within grafts, marked by expression of the common antigen CD3, and found no significant effects of graft type ($P = 0.851$;

Fig. 4d, e). All grafts contained CD3$^+$ T cells, which were present not only in the perivascular space but also distributed throughout grafts (Supplementary Fig. 3). Similar to our observations for phagocytic cells, there was a large inter-subject variation in CD3$^+$ lymphocytes within the treatment group, suggesting that other factors besides biological sex may influence the infiltration of T cells into graft tissue.

**Fig. 3 | Hypervascularization and perivascular hypercellularity in male grafts placed within female hosts. a** Representative image of blood vessels within GFP+ graft. The vasculature is visualized by CD31 immunoreactivity on vascular endothelial cells; outlines of blood vessels are drawn with dotted lines. **b** High-magnification images of grafts showing blood vessels outlined with dotted lines. **c** Quantification of the total vascular area within grafts. MM ($n = 9$ mice); MX ($n = 4$ mice); MF ($n = 9$ mice); FF ($n = 5$ mice); FX ($n = 8$ mice); FM ($n = 7$ mice). MM vs. FM: $P < 0.0001$; MX vs. FM: $P = 0.0008$; MF vs. FM: $P < 0.0001$; FF vs. FM: $P < 0.0001$; FX vs. FM: $P < 0.0001$. **d** Quantification of vascular diameter within grafts. MM ($n = 6$ mice); MX ($n = 4$ mice); MF ($n = 7$ mice); FF ($n = 5$ mice); FX ($n = 8$ mice); FM ($n = 7$ mice). MX vs. FM: $P = 0.0030$; FF vs. FM: $P = 0.0228$; FX vs. FM: $P = 0.0003$. **e** Representative images of grafts labeled with nuclear marker DAPI. Hypercellularity is evident by increased numbers of DAPI+ nuclei around blood vessels in the

female host/male graft group (asterisks). **f** Process of drawing perivascular ROIs. First, ROIs were drawn for blood vessels (left), then ROIs were enlarged by 50 μm (middle), and the number of DAPI + nuclei were quantified within enlarged perivascular ROIs (right). **g** Quantification of DAPI+ cell density within the 50 μm perivascular zone surrounding blood vessels in grafts. MM ($n = 9$ mice); MX ($n = 4$ mice); MF ($n = 9$ mice); FF ($n = 5$ mice); FX ($n = 6$ mice); FM ($n = 9$ mice). MM vs. FM: $P = 0.0058$; MX vs. FM: $P = 0.0094$; MF vs. FM: $P = 0.0022$; FF vs. FM: $P = 0.0326$; FX vs. FM: $P = 0.0011$. All data are mean ± SEM. *$P < 0.05$, **$P < 0.01$, ***$P < 0.001$, ****$P < 0.0001$ versus FM by two-way ANOVA with Sidak's multiple comparisons test. Scale bars = 100 μm (**b**), 50 μm (**a**, **e**). Source data are provided as a Source Data file. The experiments in **a**, **b**, **e**, **f** were performed twice with similar results.

Distinct subsets of T cells have different functions[48]; after SCI, they can modulate both reparative and degenerative processes in the injured spinal cord[49]. Cytotoxic CD8+ T cells have pro-degenerative functions that impede recovery after SCI[50] and traumatic brain injury[51,52]. We next quantified the numbers of infiltrating CD8+ T cells throughout the entire grafts and found that there was a significant effect of graft type on cell density ($P = 0.0013$, Fig. 4d, f). Specifically, we found that FM grafts contained significantly higher density of CD8+ cells than all other groups (Fig. 4f). When we analyzed the numbers of CD3+ and CD8+ T cells that were present only within perivascular regions, we found that FM grafts contained significantly more perivascular CD3+ cells than MM and MF grafts (Fig. 4g), and that FM grafts contained significantly more perivascular CD8+ cells than MM, MX, MF, and FF grafts (Fig. 4h). When we evaluated the ratio of CD8+/CD3+ cells in grafts, we found that $45.5 \pm 12.0\%$ of the T cells in FM grafts were immunoreactive for CD8, which was significantly higher than all other groups except for the FX group ($33.9 \pm 12.1\%$) (Fig. 4i). Hence, greater proportions of T cells in male grafts within female mice are cytotoxic, indicating that sex-mismatched grafts provoke an immune rejection response in female mice but not male mice. While not statistically significant, we observed a trend toward increased cytotoxic CD8+ cells in FX grafts (Fig. 4f, h, i); this suggesting that the presence of even ~50% male cells is sufficient to induce this response. This response is not due to differential levels of GFP expression in male and female grafts, because GFP intensity was similar in both graft types (Supplementary Fig. 4). Additionally, we found that male and female mice did not exhibit significantly different cellular responses to injury in the absence of transplanted NPCs, suggesting that the differences we observed in this study are due to the biological sex of grafts and not due to innate differences in the host immune response (Supplementary Fig. 5).

## Discussion

Neural stem- and progenitor cell transplantation is currently being evaluated in clinical trials for SCI[53,54]. While these early trials will be critical for determining safety of treatment, further characterization of NPCs may be required to improve safety and/or efficacy in future trials. Here we have shown that biological sex is a significant factor influencing the immune response of syngeneic mice to transplanted NPCs. Although the risk of sex mismatch in solid organ transplantation has been well described (Table 1), it remains unknown whether sex mismatch might also be a risk factor for graft rejection in human cell transplantation trials for SCI. Human leukocyte antigen (HLA)-matching has been thoughtfully considered in selecting the human iPSC-derived NPCs that have recently been advanced to clinical trials. Our findings suggest that potential interactions between the sex of the donor cells (the super-donor integration-free hiPSC line YZWJs513, which is derived from a male donor) and the recipient patients will be important to analyze[53,54]. More broadly, as greater numbers of cell transplantation studies for SCI and other conditions advance to clinical trials, we propose that donor cell sex should be openly reported.

Our findings also have implications for experimental SCI studies. To our knowledge, the sex of donor cells (whether rodent or human) has not previously been reported for any preclinical NPC transplantation study for SCI. For studies utilizing primary rodent cells, cells obtained from one or more litters of embryos are typically pooled together prior to transplantation[55]. In this study, we found that mouse litters were composed of equal proportions of male and female embryos on average, but some individual litters were predominantly composed of one sex (Table 2). Some strains of transgenic mice may produce litters that typically exhibit skewed sex ratios[56,57]. We speculate that batch-to-batch inconsistency in the female/male ratio of donor cells could lead to unexplained variability in outcomes such as those reported here and potentially long-term graft survival. Our findings underscore the importance of selecting the most appropriate sex of host animals for SCI cell transplantation studies. We propose that the use of male mice as host animals may be best for studies utilizing transplantation of mixed-sex NPCs, especially if immunological outcomes are to be examined.

In this study we utilized syngeneic mice, allowing us to isolate the role of biological sex on immune-related outcomes. Although we did not examine the molecular mechanisms mediating female rejection of male donor tissue, we speculate that the male-specific transplantation antigens, H-Y antigens, are implicit in this response. H-Y antigens are minor histocompatibility antigens encoded on the male Y chromosome; the epitopes are expressed on the cell surface of most male cells and are recognized by cytotoxic T cells in recipients lacking a Y chromosome[58–61]. Notably, H-Y antigens have been shown to mediate male graft rejection by female mice[58,62,63] and male organ transplant rejection by human females[38,43,44,64]. This possibility could be examined in future NPC transplantation studies, for instance by assessing the presence of H-Y-specific CD8+ T lymphocytes[65] in the circulating bloodstream of female animals following transplantation of male cells. Aside from H-Y histocompatibility issues, hormones can also play roles in sex-based graft rejection. For example, recipient estradiol levels have been shown to affect kidney transplant survival[66]. More work is needed to elucidate the mechanisms of sex-dependent graft rejection in the context of SCI and NPC transplantation.

The inflammatory response following SCI is complex, dynamic, and a subject of intensive research[67–70]. In contrast, little is known about the interactions between the host immune system and NPCs transplanted into the injured spinal cord. Here, we have shown that syngeneic NPC grafts become populated with CD68+/Iba1+ macrophages/microglia and CD3+ and CD8+ T cells. All of these cell types have been shown to be part of the immune response to SCI[35,68,71,72], but other cell types such as neutrophils are also involved[73,74]. While we did not examine these cell types in the present study, future work is needed to characterize the complexity of the inflammatory response at acute and chronic time points following transplantation of sex-mismatched cells. We also observed significantly enlarged vascular structures in male NPC grafts within female hosts. The physiological relevance of this is unclear; the mechanisms of NPC graft

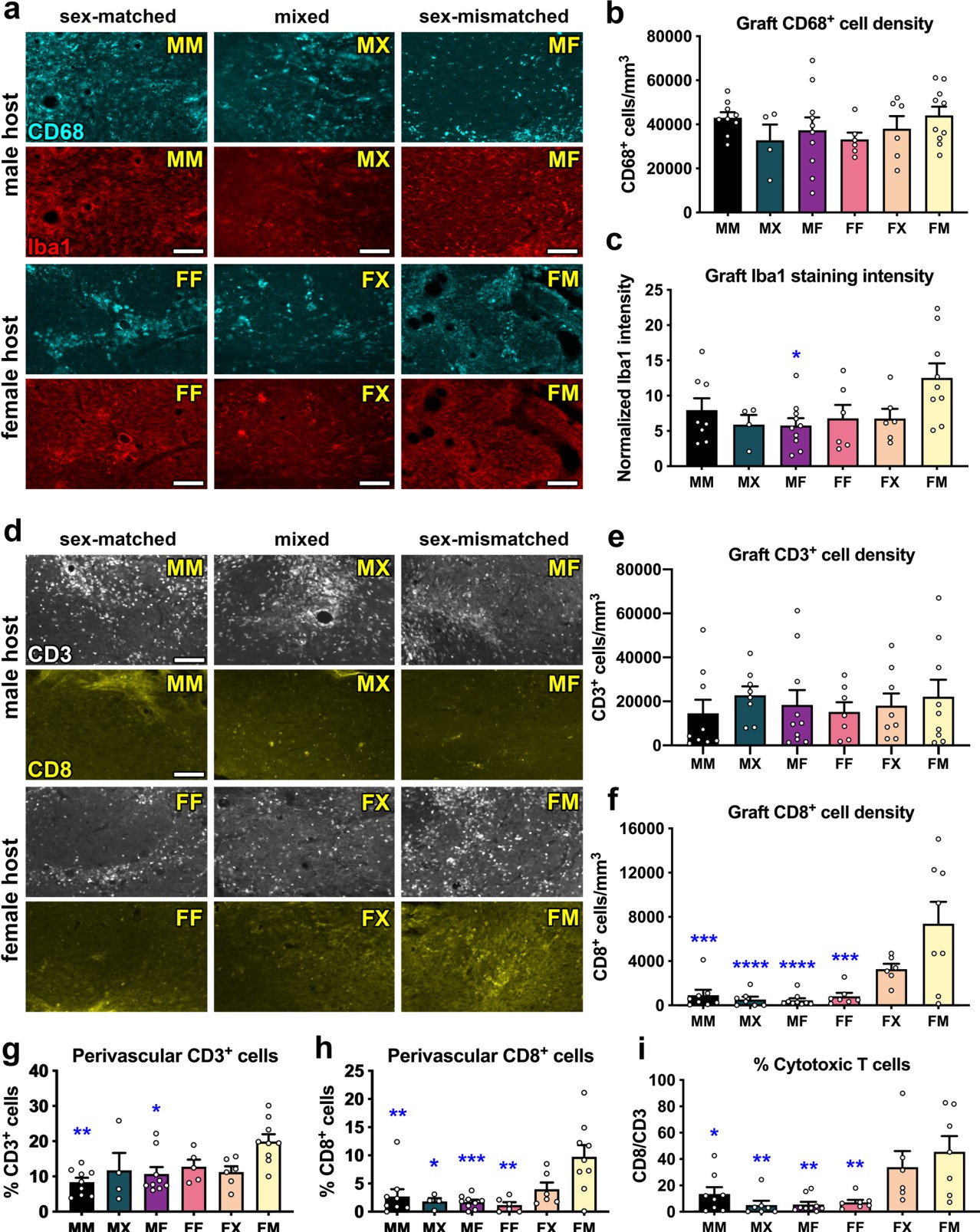

vascularization are incompletely understood, and to our knowledge there have not previously been reports of abnormal vascular morphology in grafts. The observation that these enlarged vessels are frequently associated with perivascular hypercellularity suggests that the morphology is associated with inflammation. Sroga et al. described dense perivascular infiltrates in the contused mouse spinal cord, which contained some T cells but mostly undefined cell types[35]. We made

similar observations in the current study; T cell density was highest in perivascular regions of FM grafts, but not limited to these regions (Supplementary Fig. 3). Certainly, more work is needed to characterize the interactions of immune cells with transplanted NPCs more thoroughly. Clinical efficacy of cell transplantation therapies for SCI will require further characterization of the fundamental biology of NPCs in preclinical animal models.

**Fig. 4 | Male grafts within female hosts exhibit increased cytotoxic T cell infiltration compared to other groups. a** Representative images of CD68 (top rows; cyan) and Iba1 (bottom rows; red) immunoreactivity in each treatment group. The same tissue sections were co-labeled with CD68 and Iba1, so each top and bottom panel show the same field of view. **b** Quantification of the density of CD68+ cells within grafts. MM ($n = 9$ mice); MX ($n = 7$ mice); MF ($n = 10$ mice); FF ($n = 6$ mice); FX ($n = 6$ mice); FM ($n = 10$ mice). **c** Quantification of the density of Iba1 immunoreactivity within grafts. MM ($n = 8$ mice); MX ($n = 7$ mice); MF ($n = 10$ mice); FF ($n = 6$ mice); FX ($n = 6$ mice); FM ($n = 9$ mice). MF vs. FM: $P = 0.0230$. **d** Representative images of CD3 (top rows; white) and CD8 (bottom rows; yellow) immunoreactivity in grafts. CD3 and CD8 are both labeled with rat anti-mouse antibodies, so the top and bottom rows represent different tissue sections. **e** Quantification of CD3+ cell density within grafts. MM ($n = 9$ mice); MX ($n = 8$ mice); MF ($n = 10$ mice); FF ($n = 7$ mice); FX ($n = 8$ mice); FM ($n = 9$ mice). **f** Quantification of CD8+ cell density within grafts. MM ($n = 8$ mice); MX ($n = 7$ mice); MF ($n = 9$ mice); FF ($n = 7$ mice); FX ($n = 6$ mice); FM ($n = 8$ mice). MM vs. FM: $P = 0.0001$; MX vs. FM: $P < 0.0001$; MF vs. FM: $P < 0.0001$; FF vs. FM: $P = 0.0002$. **g, h** Quantification of the % of DAPI+ cells in perivascular regions that are CD3+. MM vs. FM: $P = 0.0016$; MF vs. FM: $P = 0.0199$. MM ($n = 9$ mice); MX ($n = 4$ mice); MF ($n = 9$ mice); FF ($n = 5$ mice); FX ($n = 6$ mice); FM ($n = 9$ mice). **h** Quantification of the % of DAPI+ cells in perivascular regions that are CD8+. MM vs. FM: $P = 0.0031$; MF vs. FM: $P = 0.0006$; FF vs. FM: $P = 0.0011$; MX vs. FM: $P = 0.0127$. MM ($n = 9$ mice); MX ($n = 4$ mice); MF ($n = 9$ mice); FF ($n = 5$ mice); FX ($n = 6$ mice); FM ($n = 9$ mice). **i** Percentage of T cells that express CD8. MM ($n = 8$ mice); MX ($n = 7$ mice); MF ($n = 9$ mice); FF ($n = 7$ mice); FX ($n = 6$ mice); FM ($n = 8$ mice). MM vs. FM: $P = 0.0268$; MX vs. FM: $P = 0.0030$; MF vs. FM: $P = 0.0016$; FF vs. FM: $P = 0.0056$. All data are mean ± SEM. *$P < 0.05$, **$P < 0.01$, ***$P < 0.001$, ****$P < 0.0001$ versus FM by two-way ANOVA with Sidak's multiple comparisons test. Scale bars = 100 μm. Source data are provided as a Source Data file. The experiments in panels a and d were performed twice with similar results.

There are some limitations to our study. First, we did not characterize the exact identities of the cells contained within the cell isolate upon transplantation. In a previous study, we showed that the cell isolate obtained from E12.5 mouse spinal cords and used for grafting contained neural and glial progenitors as well as distinct dorsal/ventral progenitor phenotypes[17]. We do not anticipate that the phenotypes of the NPCs used in the present study, obtained following the same isolation protocol, are significantly different than what we previously observed; indeed, in both studies we have used primary spinal cord progenitors that were not cultured, purified, or otherwise manipulated. However, we acknowledge that more extensive characterization of the cell isolate would yield greater insight into the exact makeup of the cells used for grafting, and will be informative for future studies. The second major limitation is that we did not examine behavioral outcomes in this study. It has previously been shown that the mild SCI lesion model used here results in impaired skilled forelimb reaching and grasping function[75], and in other studies NPC transplantation has been shown to improve these functions after SCI[14,76]. However, in the present study we did not assess whether graft sex mismatch in female animals might lead to worsened functional outcomes over time compared to other groups. Beyond motor functional recovery, it is possible that graft sex mismatch could potentially affect other, long-term consequences such as graft rejection, pain, or systemic inflammation. These possibilities should be addressed in future work.

## Methods

### Ethics statement
Animal studies were performed in stringent compliance with the *NIH Guidelines for Animal Care and Use of Laboratory Animals*. All experiments utilizing animals were approved by the Texas A&M University Institutional Animal Care and Use Committee. All efforts were made to minimize pain and distress.

### Animals
A total of 90 mice were used for this study, including $n = 69$ 8–10 week-old C57BL/6 mice (#000664, Jackson Laboratories; $n = 45$ males and $n = 36$ females) and $n = 9$ 8–10 week-old adult female GFP mice [C57BL/6-Tg(CAG-EGFP)131Osb/LeySopJ; #006567, Jackson Laboratories]. Animals had free access to food and water throughout the study and were group-housed in standard Plexiglas cages on a 12-h light/12-h dark cycle (light cycle = 6:00 am–6:00 pm). Animal housing facilities had ambient temperature between 20–23 °C and 30–70% humidity. During the study, six animals died within 2 days post-surgery. This produced the final group sizes shown in Table 3.

### Spinal cord injury
The dorsal column lesion model of spinal cord injury was utilized for this study for ease of animal care, and because this SCI model supports consistent graft survival without requiring a fibrin matrix[17]. Dorsal column lesions were performed at cervical spinal cord level 4 (C4). All surgeries were performed under deep anesthesia using a combination of ketamine (25 mg/kg), xylazine (5.8 mg/kg), acepromazine (0.25 mg/kg), and inhaled isoflurane (0.5–1%). Following laminectomy at C4, a tungsten wire knife with an extruded diameter of 1.0–1.5 mm (McHugh Milieux, Downers Grove, IL) was centered above the spinal cord midline, retracted, and inserted to a depth of 0.8 mm below the dorsal spinal cord surface. The arc of the knife was then extruded and raised to transect the spinal cord dorsal columns[77].

### Neural progenitor cell dissociation and transplantation
Mouse embryos were generated through timed mating between wild-type females and GFP males. Adult female mice received intraperitoneal injections of luteinizing hormone-releasing hormone (5 I.U.; Sigma-Aldrich #L-4513) and 4 days later, females were paired with males overnight. Pregnancy was confirmed by palpation of the abdomen 12 days later. Spinal cords and limb bud tissue from E12.5 GFP mouse embryos were dissected in ice-cold Hank's Balanced Salt Solution (HBSS) on the morning of surgery and stored in individual tubes on ice for 3–4 h while genotyping was performed. Spinal cords were then pooled by sex and digested in 0.125% trypsin at 37 °C for 8–10 min (≤10 spinal cords were pooled for a single cell preparation). Fetal bovine serum (10% in Dulbecco's Modified Eagle Medium) was then added at a 10:1 volume ratio to halt the trypsinization reaction, and spinal cords were centrifuged at 600 RCF for 2 min. The supernatant was removed and tissue was gently triturated in Neurobasal Medium +2% B27 Supplement (NBM/B27) until cell suspension appeared milky and homogeneous (~15–20 passes through a P1000 pipette tip). Cell suspensions were then centrifuged at 600 RCF for 2 min. The supernatant was removed and cells were resuspended in 2–3 mL of NBM/B27, then passed through a 40-μm cell strainer. Cell viability was assessed by trypan blue exclusion and confirmed to be >95% in all cases. Cells were stored on ice in NBM/B27 until immediately prior to transplantation, then resuspended to a concentration of 300,000 viable cells/μL in HBSS. NPCs were transplanted within 30 min following spinal cord injury surgery. We chose to graft immediately

**Table 3 | Experimental groups in this study**

| Host animal sex | Grafted NPC sex | Abbreviation | N |
|---|---|---|---|
| Male | Male | MM | 9 |
| Male | Mixed | MX | 8 |
| Male | Female | MF | 10 |
| Female | Female | FF | 9 |
| Female | Mixed | FX | 8 |
| Female | Male | FM | 10 |
| Total | | | 54 |

following SCI because previous studies have demonstrated that NPC grafts survive well and integrate with host spinal cord when grafted immediately into a cervical dorsal column lesion in mice and rats[17,27]. Combining the SCI and grafting procedures into one surgical session also minimizes the number of surgical procedures performed on the experimental animals. Cells used for transplantation were comprised of either 100% male NPCs, 100% female NPCs, or 50% male/50% female (mixed) NPCs. A 1.5 µL volume of cells was injected into the lesion cavity at a depth of 0.5–0.8 mm via pulled glass micropipette using a PicoSpritzer II (General Valve, Inc., Fairfield, NJ), over a period of 5 min. Surgeries took place over 4 days, using 1–3 litters of embryos per day to obtain donor cells. Subjects were randomly assigned to receive male, female, or mixed NPCs. Donor cells of both sexes were transplanted each day, and both male and female host subjects were used each day. Following cell transplantation, incised muscles were sutured with 4–0 prolene sutures and stainless steel wound clips were used to close the incised skin. Antibiotic powder (Neo-Predef, Zoetis Inc, Kalamazoo, MI) was applied to the sutured muscles prior to skin closure. Post-operative care consisted of subcutaneous injection of banamine (0.05 mg/kg) and ampicillin (0.05 mg/kg) in lactated Ringer's (0.5 mL) once daily for 3 days, and animal cages remained half on/half off heating pads for 72 h post-surgery. Daily health checks were performed for the duration of the study in addition to monitoring animal grooming behaviors and weight.

### Genotyping

Embryonic tissue (tails and heads) was genotyped using primers directed against the *Rmb31x/Rmb31y* gene as previously described[26]. This approach allows for the rapid discrimination of male versus female genotype by simplex PCR. Briefly, primers (Forward: CACCT-TAAGAACAAGCCAATACA; Reverse: GGCTTGTCCTGAAAACATTTGG) were designed to amplify an 84-bp sequence present in *Rmb31y* but not *Rbm31x* genes, producing a 269-bp product from the X chromosome and a 353-bp product from the Y chromosome[26]. Tissue was genotyped using the Terra™ PCR Direct Genotyping Kit (Takara Bio, Inc.). The amplification product was run on a 4% agarose gel for 45 min at 90 V, and gels were visualized using GelRed nucleic acid stain (Millipore Sigma).

### Immunohistochemistry

Four weeks following SCI and NPC transplantation, subjects were euthanized by anesthesia overdose and transcardially perfused with 30 mL of 0.1 M phosphate buffer (PB) followed by 30 mL of 4% paraformaldehyde (PFA) in 0.1 M PB. Spinal columns were post-fixed in 4% PFA in 0.1 M PB overnight at 4 °C, then cryopreserved in 30% sucrose in 0.1 M PB for at least 3 days at 4 °C. Spinal cords were then removed and a 1-cm length of spinal cord centered around C4 was embedded in Tissue-Tek OCT compound (VWR) and frozen on dry ice. Spinal cord tissue was cryosectioned in the sagittal plane to a thickness of 30 µm. Sections were collected into a 24-well plate and stored at 4 °C.

Either a 1-in-6 or a 1-in-12 tissue series was used for each round of immunohistochemistry. Sections were washed in tris-buffered saline (TBS) three times for 10 min each, then blocked in tris-buffered saline (TBS) containing 5% donkey serum (Lampire Biological Laboratories, #7332100) and 0.25% Triton-X-100 (Sigma-Aldrich) for 1 h at room temperature. Sections were then incubated with primary antibodies (Supplementary Table 2) diluted in blocking solution overnight at 4 °C. The next day, sections were washed in TBS three times for 10 min each, then incubated with AlexaFluor-conjugated secondary antibodies (Jackson ImmunoResearch) diluted to a concentration of 1:1000 in blocking solution for 2 h at room temperature. Finally, sections were washed in TBS three times for 10 min each, with the final wash containing DAPI (5 µg/mL, Sigma-Aldrich, D9542). Sections were mounted to gelatin-coated slides, air-dried, rinsed in distilled water, and coverslipped with Mowiol mounting medium.

### Image acquisition

Slides stained with fluorescent dyes were stored in the dark at −20 °C and imaged in a dark room. Images were acquired using the same acquisition settings across all samples for each immunohistochemical label. Slides were imaged using a Nikon Eclipse upright fluorescent microscope equipped with a Prior Scientific XY motorized stage and a Zyla 4.2 PLUS monochrome camera (Andor). Nikon NIS-Elements software was used for image acquisition and XY stitching. Images were captured with a 10× magnification objective. To generate representative images (not used for quantification), the Extended Depth of Focus module in NIS-Elements was sometimes used to create focused images from Z-stacks. Images were exported as 8-bit TIFF files for analysis.

### Image analysis

All image analysis was performed in a blinded fashion by at least two independent experimenters using ImageJ or FIJI software. Images of GFP immunoreactivity were used to draw regions of interest around the border of graft (graft ROIs) for each individual image. Automated cell counting methods were always validated by manual counts. Samples that showed poor immunostaining were excluded from analysis. Samples for which there were no sections with clear graft tissue visible were excluded from analysis. Typically, for a given sample, a 1-in-6 series of tissue contained between 3–5 sections with GFP signal. For tissue sections of lesion-only animals, an ROI was drawn around the lesion border (defined by GFAP immunoreactivity) and expanded by 500 µm to include the lesion and perilesional area, which was used for quantification.

**NeuN quantification.** NeuN immunoreactivity within graft ROIs was thresholded using the ImageJ Auto Local Threshold function with Bernsen's thresholding method[78]. Watershed was applied to binary images and the Analyze Particles function was used to count the total number of NeuN$^+$ cells.

**Axon outgrowth quantification.** GFP fluorescence was overexposed so that fine GFP$^+$ processes were visible at sites distant from the main body of the graft. Graft ROIs were translated in 500-µm increments for 5 mm in both rostral and caudal directions. At each 500-µm increment, the total number of GFP$^+$ axons crossing the leading edge of the ROI was manually counted. Data are represented as the average number of axons per tissue section in each treatment group.

**GFAP quantification.** For each graft ROI, integrated density (the sum of the values of the pixels in the ROI) was divided by the number of pixels in the ROI to obtain the mean gray value. To account for differences in background intensity, data for each individual image were normalized to the mean gray value of GFAP immunoreactivity in the host gray matter at least 4 segments rostral to the site of injury/graft in the same image.

**Vascular quantification.** Images of CD31 immunoreactivity were used to visualize vascular structures. ROIs were manually drawn around all visible blood vessels within graft ROIs. The total vascular area of grafts was calculated by dividing the number of pixels within vascular ROIs by the number of pixels within graft ROIs. The vascular diameter was calculated by measuring the distance of lines drawn orthogonally across blood vessels in grafts that were visible longitudinally.

**DAPI quantification.** The image containing the largest cross-sectional graft area for each animal was used for analysis. The vascular ROIs for that image were enlarged by 50 µm in ImageJ, and then the original vascular ROI was subtracted from the enlarged ROI, to obtain perivascular ROIs. Using the DAPI image channel, DAPI$^+$ cells within the perivascular ROI were counted manually.

**Iba1 quantification.** For each graft ROI, integrated density (the sum of the values of the pixels in the ROI) was divided by the number of pixels in the ROI to obtain the mean gray value.

**CD3/CD8/CD68 quantification.** $CD3^+$, $CD8^+$, and $CD68^+$ cells in grafts were manually counted.

### Statistical analysis
GraphPad Prism 8 (GraphPad Software, Inc.; La Jolla, CA) was used to perform statistical analysis. Details for all statistical tests are provided in Supplementary Table 1. All data are presented as mean ± SEM. Statistical significance was defined as $P < 0.05$. All tests were two-tailed.

### Reporting summary
Further information on research design is available in the Nature Research Reporting Summary linked to this article.

## Data availability
Raw data will be made available upon request, within 2 weeks of the request. The authors declare that all data supporting the findings of this study are available within the paper and its supplementary information files "Source Data". The source data are provided as a Source Data file with this paper. Source data are provided with this paper.

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

## Acknowledgements

We thank Dr. Darrell Pilling and Dr. Dylan McCreedy for helpful scientific discussion, and Kim Loesch and Amy Leonards for research support. We gratefully acknowledge funding from the National Institutes of Health

(R01NS116404 to J.N.D., R35GM138098 to H.B.); Craig H. Neilsen Foundation (J.N.D.); Mission Connect, a project of the TIRR Foundation (J.N.D.); and Paralyzed Veterans of America Research Foundation (J.N.D.). We thank the Texas A&M University LAUNCH Undergraduate Research Scholars program for supporting M.P. during this project.

## Author contributions

M.P. performed experiments and data analysis, wrote the manuscript, and contributed to study design. M.A. performed animal surgeries and embryo generation, and contributed to study design. P.A.K. performed immunohistochemistry, image acquisition, and image analysis. G.D. performed immunohistochemistry, image acquisition, and image analysis. P.G. performed immunohistochemistry, image acquisition, and image analysis. A.T. performed NPC isolation and tissue processing. V.D. assisted with animal surgeries and performed genotyping. D.M. performed image acquisition and data analysis. S.L. performed tissue processing and immunohistochemistry. M.M.J. performed genotyping. D.B. performed data analysis. H.B. participated in study design and statistical analysis. J.N.D. designed the study, performed animal surgeries, and wrote the manuscript.

## Competing interests

The authors declare no competing interests.
