## [Peer Review File · Nature Communications]

Effects of biological sex mismatch on neural progenitor cell transplantation for spinal cord injury in miceREVIEWER COMMENTS

Reviewer #1 (Remarks to the Author):

This study examines the role of donor sex on the fate of neural progenitor cell (NPC) grafts after spinal cord injury (SCI) and reports that male grafts to female hosts exhibit significantly increased infiltration of cytotoxic T cells and significantly reduced numbers of surviving graft-derived neurons compared with sex matched grafts-host. The experimental approaches are well designed, well controlled, and appropriate. The experimental evidence provided is compelling. Experiments are properly controlled and have sufficient replicates to have statistical power. The findings do not appear over-interpreted, and I found them convincing. I think the findings are important in a broad context. Grafting of NPC in various forms is increasingly explored and considered in potential treatments for a broad variety of neurological disorders in addition to SCI. As the authors note, there is surprisingly little information available on the role of the match or mismatch of biological sex between graft and host on outcome. The study here shows a clear detrimental effect of male graft to female hosts, and this effect matches a well known observation from a long history of grafting whole organs such as kidney, heart, lung, liver etc. The authors point out that the sex of grafted NPC and the degree of graft-host match or mismatch should be reported and controlled for in future studies. I agree. I think that this study warrants the broad audience afforded by this journal.

I have only a few specific comments to be considered.

Specific comments

- 1. As the authors note, microglia and macrophages can be difficult to distinguish from one another in injured CNS tissue. Strictly speaking, it is not correct to label the graph in Figure 4B as "Macrophage density" based simply on CD68 labeling. Ramified microglia that still have extensive processes and microglial markers (P2YR12) can express high levels CD68, which is just a marker of increased phagocytic activity. The heading in Figure 4B should be changed to "Graft CD68+ cell density" to match the other headings. This change does not alter the overall interpretation of the results, but is a more accurate representation of the data.**
- 2. Strictly speaking, the data shown in Figure 4C is not Iba1+ cell density, but rather Iba1+ overall staining intensity. As per the authors' methods, the value measured was the integrated density of staining per pixel. This value could be influenced by increases in Iba1+ molecular expression that could lead to greater staining intensity. The intensity value could also be influenced by the increased branching of microglial processes that would increase mean staining intensity per pixel. Either of these things could occur without an overall change in cell density. A better heading would simply be 'Iba1+ staining intensity'. This change also does not alter the overall interpretation of the results, but is also a more accurate representation of the data.**
- 3. The authors should add to the methods a brief statement about why they chose to graft immediately after the SCI. The authors nicely explain why they chose to use the dorsal column lesion model. Similarly, they should provide a sentence or two in the grafting section on why they chose to graft "within 30 minutes following spinal cord injury surgery". This information will be useful to experts interested in comparing different types of approaches. The timing of when grafting occurs has the potential to influence host responses and a brief comment on this point may also be appropriate.**
- 4. Minor point. In the legends for figures 3 and 4, the $p < 0.0001$ is labelled with *** and not ****.**

Reviewer #2 (Remarks to the Author):

The manuscript, "Effects of biological sex mismatch on neural progenitor cell transplantation for spinal cord injury in mice", is an original research paper, which mainly used immunohistochemical methods to identify if sex influences on neural transplantation in SCI mice. This is a very important topic in transplantation study but has never addressed in SCI field. I feel very easy and comfortable when I read though the manuscript. The experiment was well designed to look at male/female or mixed embryonic NPCs grafted in different sex of animals. The methods used were appropriate and data were technically sound. The results included that biological sex mismatch of donor cells did not affect astrocyte density, graft-derived axon outgrowth, graft macrophage/microglia density, or overall T lymphocyte infiltration. However, the lower density of NeuN-labeled neurons was observed in male grafts within female recipients, which also had significantly greater vascular diameter and perivascular hypercellularity, as well as higher degree of cytotoxic T cell infiltration. The results provide sufficient evidence for the conclusions that H-Y histocompatibility is a significant factor affecting the efficacy of NPC transplantation. I perceive rigors and transparency throughout the manuscript. The language is authentic and writing is clear and concise.

The reviewer assures it a high-quality work and data reported are robust and solid. Authors combined necessary explanations when presenting the data, making them rational and understandable. Interpretation was straightforward and reasonable without overstatement. With increasing numbers of research in cell grafting for SCI, the finding in this study is vital and unique. It is so important in this field and will definitely advance understanding in a way that moves the field forward. The reviewer has only minor concerns to be addressed.

1. In the Introduction line 94-102, The authors may consider not presenting results here. However, it is only a suggestion. If keeps as is, also fine.
2. It seems unclear how many spinal cord sections were used for quantification in each animal. If GFP signal was not in all sections in one serial of sagittal sections, it might include 3-4 sections per mouse.
3. A one-way ANOVA was used for statistical analysis in most experiments. Because both grafts and recipients used were in different sex, a two-way ANOVA might be more appropriate due to the two variables.

Reviewer #3 (Remarks to the Author):

The manuscript by Pitonak et al. describes their findings on the effects of sex mismatch on transplantation of neural progenitor cells (NPCs) following spinal cord injury (SCI). The group finds that female mice receiving male cells exhibit significantly reduced neuronal density, hypervascularization, perivascular hypercellularity, and increased numbers of cytotoxic CD8+ T-cells infiltrating the graft. The results of the study suggest that biological sex mismatch can affect transplantation outcomes in SCI. The findings have broader implications, indicating a need for consideration and reporting of biological sex of donors and recipients in SCI cell transplantation research and clinical trials. However, the current study is limited to histological analysis of transplantation and there is no assessment of functional recovery in the mice following SCI and NPC transplantation. It is unknown from the study whether the observed histological differences between experimental groups leads to any differences in locomotor ability during SCI recovery and there is no mechanism for why only male donor to female recipient mismatch results in histological differences. Only a single time point of 4 weeks post SCI is assessed, making it difficult to determine the acute versus chronic effects of biological sex mismatch on NPC transplantation. There is also no assessment of remyelination and whether sex mismatch leads to any differences between groups. The manuscript is well written, but there is no assessment of functional recovery and no attempt to look at any mechanism that may explain the results or the use of more modern molecular techniques such as single cell nuc seq to establish which cells are involved in this process.

Major points

1. This study is missing an important control group. All comparisons (except in Figure 2B where an uninjured animal is included) are made only in transplanted animals. An injured, non-transplanted control group of each sex needs to be included.
2. No functional consequences of the spinal cord injury are included. Analysis of a functional/behavioral readout should have been conducted. In the absence of this, it is impossible to determine whether the anatomical differences observed in females receiving male transplants matter in terms of functional outcomes.
3. The cells being transplanted are not characterized, nor does there appear any sort of quality control to ensure the isolations resulted in the desired cell type. According to the methods, spinal cords were isolated from E12.5 pups, dissociated, and then transplanted, without characterizing what cell types are contained in the isolate.
4. The authors state in the last sentence of the introduction that "this work suggests that H-Y histocompatibility may be a significant factor affecting the efficacy of NPC transplantation." However, studies to address the potential mechanism underlying observed group differences are not conducted. Moreover, in the discussion the authors postulate that H-Y antigens are responsible for the observed effects and state that this could be examined by looking for H-Y specific CD8+ T cells in the bloodstream. Was blood collected from the animals in the current study? This assay would help elucidate the mechanism more clearly and significantly increase the conclusions that could be drawn from this study. A discussion of other potential mechanisms should also be included.
5. Based on Supplementary table 1, while each group has at least 8 animals, the number that are actually analyzed differs considerably across the different IHC outcomes and is often only 4 animals for a particular group. Why are all 8 animals not included in these analyses? This is especially important in the case of Figure 4B and C, where there is no difference in CD68+ cells within grafts, but IBA1+ cells are increased in the FM group. This reaches significance only in the comparison between FM and MF groups but FM appears to be higher than other groups in general. The two that are significant have more animals than any of the other groups (9 and 10 as compared to 4, 6, and 8) so if more n's were added to the other groups, these might be statistically significantly different from FM as well. The authors should stain for and quantify IBA1 in the remaining animals to determine if the FM group does in fact have increased microglial infiltration compared to all the other groups.
6. In Figure 2B, there is no significant main effect and yet the authors go on to report that the FM group is significantly different from the uninjured group. The absence of a significant main effect precludes post-hoc comparisons and thus, this should not be reported as significant, as it is misleading.
7. In Figure 2F, there are three sites where males receiving male cells (MM) are significantly different from females receiving male cells (FM). Based on the graph the MM have more GFP+ axons than the FM group. The authors do not go into this and instead end with saying that sex of host and donor do not affect axon outgrowth; however, at least in three locations, they actually do.
8. In Figure 4, the goal is to identify the cell type found in perivascular infiltrates. The authors stain for various immune markers but then end up analyzing their expression within the entire graft site, not specifically within the vascular areas that they see increased in the FM group.
9. In Figures 4D-E, CD3 is used to identify T cells, finding no difference among groups. The authors state that CD3+ T cells were found throughout the graft, not only in the perivascular space. However, the largest clusters of these cells do appear to be around vessels while the rest of the tissue shows rather sparse staining, as shown in Supplementary Figure 3.
10. In Figures 4F-G, cytotoxic CD8+ T cells as well as the ratio of CD8 to CD3 T cells were increased in FM grafts. Interestingly, these measures were also increased in female mice receiving mixed donor cells, suggesting that the presence of even 50% male cells induced an immune rejection response in females. This should be noted and discussed.

11. In discussing hypercellularity observed in Figure 3E-G the authors reference Supplemental Fig 3 which is of a different stain entirely and relates to T cell infiltration

12. Finally the authors should discuss the use of GFP as a marker of the cells which in itself may be immunogenic. perhaps the male animals expressed a higher level of GFP? was this explored?.

We thank the reviewers for their time and effort in reviewing our manuscript. We appreciate the strong positive feedback about our study, and we agree with the reviewers that our findings are important in a broad context. We have carefully considered all of the reviewers' comments and integrated suggested changes into the revised version of the manuscript. Please see below for a point-by-point response to reviewers' comments (our responses are in blue). Note that we have highlighted changes from the original version in the revised manuscript.

Reviewer #1:

1. As the authors note, microglia and macrophages can be difficult to distinguish from one another in injured CNS tissue. Strictly speaking, it is not correct to label the graph in Figure 4B as "Macrophage density" based simply on CD68 labeling. Ramified microglia that still have extensive processes and microglial markers (P2YR12) can express high levels CD68, which is just a marker of increased phagocytic activity. The heading in Figure 4B should be changed to "Graft CD68+ cell density" to match the other headings. This change does not alter the overall interpretation of the results, but is a more accurate representation of the data.

The reviewer is correct. We have made this change, and the heading for panel 4B is now "Graft CD68+ cell density".

2. Strictly speaking, the data shown in Figure 4C is not Iba1+ cell density, but rather Iba1+ overall staining intensity. As per the authors' methods, the value measured was the integrated density of staining per pixel. This value could be influenced by increases in Iba1+ molecular expression that could lead to greater staining intensity. The intensity value could also be influenced by the increased branching of microglial processes that would increase mean staining intensity per pixel. Either of these things could occur without an overall change in cell density. A better heading would simply be 'Iba1+ staining intensity'. This change also does not alter the overall interpretation of the results, but is also a more accurate representation of the data.

We have made this correction, and the heading for panel 4C is now "Graft Iba1 staining intensity".

3. The authors should add to the methods a brief statement about why they chose to graft immediately after the SCI. The authors nicely explain why they chose to use the dorsal column lesion model. Similarly, they should provide a sentence or two in the grafting section on why they chose to graft "within 30 minutes following spinal cord injury surgery". This information will be useful to experts interested in comparing different types of approaches. The timing of when grafting occurs has the potential to influence host responses and a brief comment on this point may also be appropriate.

Great point. We have updated the Methods section (page 11, lines 337-341) with the following sentences: "*We chose to graft immediately following SCI because previous studies have demonstrated that NPC grafts survive well and integrate with host spinal cord when grafted immediately into a cervical dorsal column lesion in mice and rats (Adler et al., 2017; Dulin et al., 2018). Combining the SCI and grafting procedures into one surgical session also minimizes the*

number of surgical procedures performed on the experimental animals.”

4. Minor point. In the legends for figures 3 and 4, the $p < 0.0001$ is labelled with *** and not ****. Thanks for catching this mistake; we have corrected both instances.

Reviewer #2:

1. In the Introduction line 94-102, The authors may consider not presenting results here. However, it is only a suggestion. If keeps as is, also fine.

At the reviewer's suggestion, we have deleted these lines from the Introduction.

2. It seems unclear how many spinal cord sections were used for quantification in each animal. If GFP signal was not in all sections in one serial of sagittal sections, it might include 3-4 sections per mouse.

This is correct – because the graft is present only in the midline of the spinal cord in most cases, GFP signal was usually only present in 3-5 sections per mouse, depending on the medial/lateral extent of the graft. We added the following sentence in the Methods section (page 13, lines 407-411) to clarify this: “Typically, for a given sample, a 1-in-6 series of tissue contained between 3-5 sections with GFP signal.”

3. A one-way ANOVA was used for statistical analysis in most experiments. Because both grafts and recipients used were in different sex, a two-way ANOVA might be more appropriate due to the two variables.

We agree with the reviewer and we have now analyzed data using a two-way ANOVA in cases when multiple graft types were compared, including data in Figures 2D, 2E, 3C, 3D, 3G, 4B, 4C, 4E, 4F, 4G. This did not change the overall conclusions of the study; in all but one case the significant differences remained the same and p-values changed only very slightly. The only difference is that in panel 4F (graft CD8+ cell density), the alpha for the FX vs. FM comparison went from significant ($p=0.0472$) to non-significant ($p=0.0647$). We have updated the manuscript text, supplement, and figures to reflect the new analysis. Please note that we did not use a two-way ANOVA to compare groups for data in 2B (NeuN density) because all groups were compared to a control (host gray matter) condition in that instance.

Reviewer #3:

1. This study is missing an important control group. All comparisons (except in Figure 2B where an uninjured animal is included) are made only in transplanted animals. An injured, non-transplanted control group of each sex needs to be included.

We agree with the reviewer that this is an important control group. We have now performed additional experiments to compare T-cell infiltration, GFAP immunoreactivity, and phagocytic cell density in the lesion sites of injury-only male and female mice (N=6 per group). We have observed no significant differences between males and females in any of these outcomes. This data is now included in the revised manuscript as Figure S5 (page 5, lines 219-223).

2. No functional consequences of the spinal cord injury are included. Analysis of a functional/behavioral readout should have been conducted. In the absence of this, it is

impossible to determine whether the anatomical differences observed in females receiving male transplants matter in terms of functional outcomes.

We thank the reviewer for their comment, but we respectfully disagree that analysis of a functional/behavioral readout should have been included in our study. We submit that functional assessments are beyond the scope of the current study. As recognized by both Reviewer #1 and Reviewer #2, our study focuses on the fundamental biology of NPC graft interaction with host, and as such it is important in a broad context. Our goal is to understand the cell biology of how NPC grafts influence the host immune response following transplantation into the spinal cord. We respectfully argue that the effects of graft rejection may not necessarily impact functional recovery (particularly in the mild SCI lesion model used in our study), but could potentially affect other, long-term consequences such as graft rejection, pain, systemic inflammation, or other undesirable outcomes. Assessing these outcomes is well beyond the scope of our study, but can be addressed in future work.

3. The cells being transplanted are not characterized, nor does there to appear any sort of quality control to ensure the isolations resulted in the desired cell type. According to the methods, spinal cords were isolated from E12.5 pups, dissociated, and then transplanted, without characterizing what cell types are contained in the isolate.

We respectfully submit that the transplanted cells used in this study (primary fetal rodent spinal cord NPCs) have been extensively characterized by our group and others. For example, in a previous study (Dulin et al., *Nat. Comm.* 2018) we characterized the cell types contained in the cell isolate as well as the mature grafts. In agreement with previous studies, we found that the isolate contained neural and glial progenitors as well as markers for specific dorsal/ventral progenitor phenotypes. In multiple studies, the mature grafts have also been characterized and found to contain spinal cord neuronal subtypes as well as macroglia. This is expected for primary spinal cord progenitors that are not cultured, purified, or otherwise manipulated. We would absolutely view characterization of the cell isolate as a necessary if, for example, we were using NPCs derived from human pluripotent stem cells, because directed/induced differentiation processes vary and the resulting cell types can be variable. We performed quality control by ensuring that all of our grafts contained neurons and glial cells, as expected (Figure 1).

4. The authors state in the last sentence of the introduction that “this work suggests that H-Y histocompatibility may be a significant factor affecting the efficacy of NPC transplantation.” However, studies to address the potential mechanism underlying observed group differences are not conducted. Moreover, in the discussion the authors postulate that H-Y antigens are responsible for the observed effects and state that this could be examined by looking for H-Y specific CD8+ T cells in the bloodstream. Was blood collected from the animals in the current study? This assay would help elucidate the mechanism more clearly and significantly increase the conclusions that could be drawn from this study. A discussion of other potential mechanisms should also be included.

The reviewer is correct that we did not conduct mechanistic studies to determine whether H-Y antigens are differentially expressed between treatment groups. Unfortunately, we did not collect blood samples from the experimental subjects so we are unable to perform additional

assays. To address the reviewer's suggestion, we have added a discussion of other potential mechanisms in the Discussion section (page 9, lines 265-268).

5. Based on Supplementary table 1, while each group has at least 8 animals, the number that are actually analyzed differs considerably across the different IHC outcomes and is often only 4 animals for a particular group. Why are all 8 animals not included in these analyses? This is especially important in the case of Figure 4B and C, where there is no difference in CD68+ cells within grafts, but IBA1+ cells are increased in the FM group. This reaches significance only in the comparison between FM and MF groups but FM appears to be higher than other groups in general. The two that are significant have more animals than any of the other groups (9 and 10 as compared to 4, 6, and 8) so if more n's were added to the other groups, these might be statistically significantly different from FM as well. The authors should stain for and quantify IBA1 in the remaining animals to determine if the FM group does in fact have increased microglial infiltration compared to all the other groups.

We thank the reviewer for this suggestion. We had originally excluded some animals from analysis due to poor immunostaining quality. We have now performed additional quantification for 3 MX samples for Iba1 and CD68 quantification, bringing the group size from N=4 to N=7. This does not change the conclusions for Fig. 4B and 4C. Unfortunately, no other tissue was remaining for additional immunohistochemistry for the other experimental groups. Therefore, we are unable to perform additional quantification.

6. In Figure 2B, there is no significant main effect and yet the authors go on to report that the FM group is significantly different from the uninjured group. The absence of a significant main effect precludes post-hoc comparisons and thus, this should not be reported as significant, as it is misleading.

We thank the reviewer for this correction, and we have revised the text and Figure 2.

7. In Figure 2F, there are three sites where males receiving male cells (MM) are significantly different from females receiving male cells (FM). Based on the graph the MM have more GFP+ axons than the FM group. The authors do not go into this and instead end with saying that sex of host and donor do not affect axon outgrowth; however, at least in three locations, they actually do.

We have now added this to the text (page 5, lines 139-141).

8. In Figure 4, the goal is to identify the cell type found in perivascular infiltrates. The authors stain for various immune markers but then end up analyzing their expression within the entire graft site, not specifically within the vascular areas that they see increased in the FM group.

9. In Figures 4D-E, CD3 is used to identify T cells, finding no difference among groups. The authors state that CD3+ T cells were found throughout the graft, not only in the perivascular space. However, the largest clusters of these cells do appear to be around vessels while the rest of the tissue shows rather sparse staining, as shown in Supplementary Figure 3.

We thank the reviewer for these comments. We had originally quantified all CD3+ and CD8+ cells throughout the graft because we wanted to characterize T cell infiltration into grafts as a whole, not simply within perivascular areas. We have now quantified the percentage of DAPI+ cells that express CD8 and CD3 in perivascular areas (included in Figure 4G-H). Notably, we

found that FM grafts had significantly greater T cell density in perivascular regions, so we have updated the conclusions to reflect this finding.

10. In Figures 4F-G, cytotoxic CD8+ T cells as well as the ratio of CD8 to CD3 T cells were increased in FM grafts. Interestingly, these measures were also increased in female mice receiving mixed donor cells, suggesting that the presence of even 50% male cells induced an immune rejection response in females. This should be noted and discussed.

We have now noted this (page 7, lines 215-218).

11. In discussing hypercellularity observed in Figure 3E-G the authors reference Supplemental Fig 3 which is of a different stain entirely and relates to T cell infiltration

We have now removed this reference to Figure S3.

12. Finally the authors should discuss the use of GFP as a marker of the cells which in itself may be immunogenic. perhaps the male animals expressed a higher level of GFP? was this explored?.

This is a good point. We have performed a quantification of GFP immunoreactivity in male and female grafts, and did not detect any significant difference ($p=0.229$, Fig. S4). We have highlighted this in the Results section (page 7, lines 218-219).

REVIEWERS' COMMENTS

Reviewer #1 (Remarks to the Author):

The authors have appropriately dealt with my concerns and as far as I can tell have also addressed the concerns of the other reviewers. The additions and edits made are appropriate and improve the paper. I have no additional comments or concerns. I think the paper is very strong and I continue to find the paper important and of high and broad interest.

Reviewer #2 (Remarks to the Author):

The reviewer's previous concerns have been adequately addressed. This updated version is substantially improved.

Reviewer #3 (Remarks to the Author):

The authors have addressed many previously noted concerns, including the analysis of injured, non-transplanted spinal cords from males and females, a statistical analysis using two-way ANOVA, quantification of additional animals for IBA1 and CD68, as well as quantification of CD3 and CD8 in the perivascular space and GFP immunoreactivity in grafts from both sexes. However, they have not addressed two major concerns: 1) the lack of functional outcomes and 2) characterization of the transplanted cells was not performed. Because additional animals would need to be included in order to analyze behavioral outcomes and this would require essentially replicating the entire study, this may not be feasible. However, these important caveats should be noted in the discussion. While in the "old days" stating E12 tissues was acceptable - newer methods of determining exactly what is in these cultures are changing the field and need to be considered. Characterizing the transplanted cells would be straightforward and would be useful in order to give readers context as to what percentage of cells are neuronal versus glial. So update to discussion regarding the limitation of not knowing exact input cells and that functional outcomes were not assessed would be needed.

We once again thank the reviewers for their time and effort in reviewing our manuscript. We are pleased that our previous revisions have sufficiently addressed the concerns of Reviewers #1 and #2. Below, we now provide a point-by-point response to the remaining concerns of Reviewer #3. Please note that we have **highlighted** changes from the previous version in the revised manuscript.

Reviewer #3:

The authors have addressed many previously noted concerns, including the analysis of injured, non-transplanted spinal cords from males and females, a statistical analysis using two-way ANOVA, quantification of additional animals for IBA1 and CD68, as well as quantification of CD3 and CD8 in the perivascular space and GFP immunoreactivity in grafts from both sexes. However, they have not addressed two major concerns: 1) the lack of functional outcomes and 2) characterization of the transplanted cells was not performed. Because additional animals would need to be included in order to analyze behavioral outcomes and this would require essentially replicating the entire study, this may not be feasible. However, these important caveats should be noted in the discussion. While in the "old days" stating E12 tissues was acceptable - newer methods of determining exactly what is in these cultures are changing the field and need to be considered. Characterizing the transplanted cells would be straightforward and would be useful in order to give readers context as to what percentage of cells are neuronal versus glial. So update to discussion regarding the limitation of not knowing exact input cells and that functional outcomes were not assessed would be needed.

We thank the Reviewer for acknowledging that we have addressed many of the previous concerns. In response to the reviewer's request, we have added a new paragraph to the Discussion section highlighting these limitations:

"There are some limitations to our study. First, we did not characterize the exact identities of the cells contained within the cell isolate upon transplantation. In a previous study, we showed that the cell isolate obtained from E12.5 mouse spinal cords and used for grafting contained neural and glial progenitors as well as distinct dorsal/ventral progenitor phenotypes¹⁷. We do not anticipate that the phenotypes of the NPCs used in the present study, obtained following the same isolation protocol, are significantly different than what we previously observed; indeed, in both studies we have used primary spinal cord progenitors that were not cultured, purified, or otherwise manipulated. However, we acknowledge that more extensive characterization of the cell isolate would yield greater insight into the exact makeup of the cells used for grafting, and will be informative for future studies. The second major limitation is that we did not examine behavioral outcomes in this study. It has previously been shown that the mild SCI lesion model used here results in impaired skilled forelimb reaching and grasping function⁷⁵, and in other studies NPC transplantation has been shown to improve these functions after SCI^{14,76}. However, in the present study we did not assess whether graft sex mismatch in female animals might lead to worsened functional outcomes over time compared to other groups. Beyond motor functional recovery, it is possible that graft sex mismatch could potentially affect other, long-term consequences such as graft rejection, pain, or systemic inflammation. These possibilities should be addressed in future work."